# Beam Search Algorithm for Anti-Collision Trajectory Planning for Many-to-Many Encounter Situations with Autonomous Surface Vehicles

**DOI:** 10.3390/s20154115

**Published:** 2020-07-24

**Authors:** Jolanta Koszelew, Joanna Karbowska-Chilinska, Krzysztof Ostrowski, Piotr Kuczyński, Eric Kulbiej, Piotr Wołejsza

**Affiliations:** 1Faculty of Computer Science, Bialystok University of Technology, 15-351 Bialystok, Poland; j.karbowska@pb.edu.pl (J.K.-C.); k.ostrowski@pb.edu.pl (K.O.); 2Unity Developer, The Dust, 50-043 Wrocław, Poland; keluthaz@gmail.com; 3R&D Department, Sup4Nav sp. z o.o., 71-602 Szczecin, Poland; eric.kulbiej@sup4nav.com; 4Faculty of Computer Science and Telecommunication, Maritime University of Szczecin, 70-500 Szczecin, Poland; piotr@am.szczecin.pl

**Keywords:** anti-collision trajectories, many-to-many encounter situation, autonomous surface vehicle, beam search algorithm (BSA), multi-surface vehicle beam search algorithm (MBSA)

## Abstract

A single anti-collision trajectory generation problem for an “own” vessel only is significantly different from the challenge of generating a whole set of safe trajectories for multi-surface vehicle encounter situations in the open sea. Effective solutions for such problems are needed these days, as we are entering the era of autonomous ships. The article specifies the problem of anti-collision trajectory planning in many-to-many encounter situations. The proposed original multi-surface vehicle beam search algorithm (MBSA), based on the beam search strategy, solves the problem. The general idea of the MBSA involves the application of a solution for one-to-many encounter situations (using the beam search algorithm, BSA), which was tested on real automated radar plotting aid (ARPA) and automatic identification system (AIS) data. The test results for the MBSA were from simulated data, which are discussed in the final part. The article specifies the problem of anti-collision trajectory planning in many-to-many encounter situations involving moving autonomous surface vehicles, excluding Collision Regulations (COLREGs) and vehicle dynamics.

## 1. Introduction

The era of autonomous ships has already begun in maritime transport. The 30-year forecast for the development of marine technologies predicts many autonomous vessels at sea [1]. This will necessitate radical implementation of new intelligent maritime navigation systems. Such systems should ensure the safety and optimal operation of the entire fleet in multi-ship encounter situations in the open sea, instead of generating a single anti-collision trajectory for each vessel. The vision of a system that manages a group of autonomous vessels was developed during the MUNIN project [2]. MUNIN was one of the first concepts of autonomous shipping, while the Milli-Ampere, Ballstad, and Autonomous Ship Transport at Trondheimsfjorden [3,4,5] projects focused on short-sea shipping—autonomous ferry shipping in particular. The Hull-to-Hull [6] project is dedicated to close proximity navigation of autonomous vessels, while Autoship [7] focuses on target detection. Yara Birkeland [8] and Advanced Autonomous Waterborne Applications [9] are dedicated to designing and building full-size autonomous ships, i.e., ships with an operating system able to make decisions and determine actions by itself. Autonomous Vessel with Air Look (AVAL) [10] is one of the first projects to use drone technology for the detection of targets at sea, while Hydrodron [11], developed by the Polish company Marine Technology, is used for autonomous hydrographic surveying in port areas, roadsteads, anchorages, lagoons, bays, lakes, rivers, and other narrow areas. The Faculty of Ocean Engineering and Ship Technology, Gdansk University of Technology, is a leading research center, executing projects Morswin and Albatross [12], which focus on underwater works and collision avoidance in 3D. The abovementioned forecast also predicts that many vessels will continue to be manned, despite the rapid growth of autonomous shipping. This makes the challenge of effective vessel traffic management even more problematic. 

Many papers are dedicated to automatic generation of a single anti-collision trajectory in one-to-one encounter situations, as well as solutions to one-to-many encounter situations. One-to-one means that only two vessels that have violated the assumed safety closest point of approach (CPA) are considered in the encounter situation. One-to-one solutions are generated by navigators as part of daily routines. If required and feasible, such solutions may be extended for one-to-many cases. However, human ability to calculate a safe trajectory accounting for many vessels is limited. Even the COLREGs suggest “dividing” a complicated situation into many one-to-one encounters. One-to-many solutions are indirectly implemented in existing collision avoidance devices, such as ARPA. However, the suggested solution is not a multistage trajectory but a new safety course. Typical solutions with multistage trajectories are delivered by some R&D projects (e.g., AVAL). A similar solution was presented in [13]. Those authors propose the beam search algorithm (BSA), which is able to determine a group of shortest paths for the “own” surface vehicle. The risk of collision between two surface vehicles, “own” and “enemy”, is calculated based on the distance path and the closest point of approach of said ships. BSA undertakes the alteration of the course and keeps the velocity constant. The ship turns to port and starboard in order to avoid the collision are used to generate a tree of possible paths. At each step, the algorithm expands those trajectories by adding consecutive maneuvers and keeping only the most promising paths. The mentioned update is executed after every reception of the novel AIS or ARPA string of data. BSA was tested with good results in real conditions on a Polish ferry, *m*/*f* Wolin. As BSA can handle up to 30 ships, it can be successfully used in congested and restricted waters.

Various methods and algorithms proposed in the literature can be classified into one of two groups of algorithms: deterministic and heuristic. Contemporary methods that generate deterministic anti-collision paths mainly try to achieve the best solution, yet in complex scenarios they tend to suffer from high computational time. The cooperative path planning algorithm [14] determines trajectories for all vessels involved in the scenario in so-called cooperative mode. Furthermore, in the trajectory-based algorithm [15], a set of paths is queried to locate the shortest of these not leading to collision and complying with the rules of the road. NAVDEC, a comprehensive navigation decision support system, implements a deterministic method [16]. Its anti-collision actions are calculated analytically, taking into account collision regulations and good seamanship. NAVDEC is a commercial software installed in many vessels and is also a subject to Polish Register of Shipping certification.

Among heuristic methods are ant colony algorithms, genetic algorithms, and particle swarm optimization. An evolutionary algorithm was introduced in [17], which produces a nearly optimal group of safe trajectories, all complying to COLREGs. A method using AIS and electronic chart display and information system (ECDIS) data and utilizing a decision support system was proposed by Tsou [18]. The evolutionary method was implemented to detect the risk of collision and to generate an optimal path. Tsau [19] and Lazarowska [20] proposed implementation of the ant colony algorithm into a decision support system designed for vessel route planning. Kang [21] used particle swarm optimization in order to generate a safe trajectory, complimenting the rules of the road. There are numerous other scientific approaches, including artificial neural networks [22], cooperative multiperson positional modelling games [23], artificial potential fields (APF) [24], or the visibility graph method [25]. The fast marching method is used in [26] in a path planner with multiple layers for an unmanned ship. The planner considers sea environments enriched with dynamic and coastal obstacles. APF, described in [24,27], is presented in order to locally optimize the path planning and dynamic obstacle avoidance tasks. APF refinement is a path-guided hybrid artificial potential field (PGHAPF) method, which is a fusion of the potential field and gradient methods [28]. The method can avoid static obstacles of any shape. In addition, the deterministic algorithm uses PGHAPF to generate anti-collision trajectories for the “own” ship under COLREGs. Furthermore, in [28] a simple waypoint selection scheme is activated together with the PGHAPF method to obtain amore optimized anti-collision trajectories in very short computational time. The algorithm, as presented in the article, is effective in restricted waters with a limited number (4) of targets (ships). It would be worth verifying the algorithm with a greater number of targets in the limited area. The area presented in the article (12 by 12 Nm) gives many opportunities to avoid a collision.

It should be mentioned that the ship’s domain method for handling optimal trajectory planning problems [29,30] is also a relevant approach. The domain plays crucial role in the process of assessing the risk of collision and deciding on a collision avoidance maneuver. Historically, the first domain was a pure circle drawn around a ship’s position, where the radius would be distance at the closest point of approach (DCPA). Fujii [31] was the first to introduce an elliptic-shaped domain. Recent studies have focused on the ship’s domain shape, as well as the source of the received border. The statistical approach is gaining major ground, as it provides insight into the actual process of imagining a ship’s domain by professional navigators in both open [32] and confined waters [33].

In [34], the authors proposed the concept of determining trajectories of ships in multi-ship encounters. To determine the ship traffic trajectories, an evolutionary algorithm is used as a global optimization tool. It finds a suboptimal solution, which is usually close to the optimal solution, in a much shorter time than when using classic optimization methods. The possibilities of its application in open and restricted waters were presented. This concept was extended in [35] to include areas where a traffic separation scheme (TSS) was established. The regulations in force in these areas were taken into account. On the other hand, in [36], based on previous research results, a skeleton method for optimizing autonomous ship trajectories was proposed for multi-ship encounters. In [37], a simple multi-ship collision situation was presented (trajectories for four ships). The solution, which is based on the deep reinforcement learning (DRL) algorithm, suggests that all four ships should take avoidance actions. However, maneuvers by two neighboring vessels could also be effective. If the actions of two vessels could be effective, is the maneuvering of four ships justified and optimal? How will the DRL algorithm work in more complex situations?

The above review of the scientific literature shows that the problems regarding generation of safe and optimal paths in ship navigation are still unsolved. Some of algorithms mentioned in the previous section have been implemented and tested as a part of a decision support system [10,37,38].

In the authors’ opinion, original solutions are needed for autonomous and non-autonomous groups of surface vehicles, not just for individual surface vehicles. Some supporting arguments are set forth below.

Argument No. 1:

Consideration of only one-to-many encounters ends up with each vessel having its own independent trajectory, which may not be safe for other ships. For example, the solution presented in [13] solves the collision situation in the master–slave system (one to many), i.e., “own” ship develops and proceeds along an anti-collision trajectory, assuming that the other ships maintain their course and speed. When these parameters change, it modifies its own trajectory again. This solution is not optimal in terms of the distance travelled and requires frequent modifications of the route.

In this situation, it is necessary to carry out an autonomous negotiation [10], which may also fail to produce safe solutions. Moreover, negotiating algorithms are time-consuming and sometimes do not finish with a consensus. While negotiations between two ships are possible in real time, allowing the implementation of agreements by both ships, the process of reconciliation between a larger number of ships is much more complex. This applies to both negotiation strategies and the conditions for finalizing negotiations. The result is an extension of the time needed to conduct negotiations, which in practice prevents the implementation of arrangements.

Argument No. 2:

While negotiations between two ships are possible in real-time, enabling the implementation of agreements by both ships, the process of reconciliation between a larger number of ships is much more complex. This applies to both negotiation strategies and the conditions for finalizing negotiations. As with the first argument, the time for negotiations is extended, preventing the implementation of arrangements. If the master–slave model is unavoidable in some situations, we can consider it as the primary solution. Of course, the master selection process is a separate challenge, which requires further research. However, if considered as complete, we can proceed with the many-to-many model.

Argument No. 3:

The use of a one-to-many solution compared to the NAVDEC system [39] has allowed shortening the distance travelled by 37%, which is the case of the daily route from Świnoujście to Trelleborg of *m*/*f* Wolin (14 maneuvers to avoid collisions per day). The results were published at the end of 2019 in *Sensors* [13]. Relating the results to a medium-sized container ship (8000 TEU), which burns around 260 tons of fuel per day when running full ahead, and assuming that each anti-collision maneuver will last only a minute less, it can save over 800 tons of fuel per year. The current (i.e., 13 November 2019) average global price of one ton of heavy fuel (HFO) is USD 406.50, which translates into savings of over USD 300,000, with more savings from an environmental point of view. These savings can be almost doubled when ships use marine diesel oil (MDO), which is required in some regions of the world. If we multiply these savings by the number of ships in the world (over 80,000), the global savings are impressive.

Such a global solution could be hard to accept from the viewpoint of a single vessel, particularly one that has to take substantive maneuvers when compared with other vessels. The awareness that one major maneuver of one’s own ship can, in a specific situation, reduce the total route of many ships will certainly facilitate the implementation of such a solution. Vessel traffic systems (VTS) can help in such an introduction and could be the first testing area for such solutions.

Argument No. 4

Modern vessel traffic systems (VTS)or vessel traffic management system (VTMS) solutions such as Wartsila [40] deliver scheduling for approaching vessels, which optimizes port loads and increases efficiency. A VTMS could also deliver a solution for a group of vessels in an area covered by the service. A proposal based on many-to-many strategies could be calculated ashore and transferred to all vessels as a recommended solution, increasing the efficiency of the whole shipping operation. Such a service will prove beneficial for ship-to-ship interactions, principally for autonomous and unmanned ships [41]. The availability of navigational intel in electronic format should diminish probable errors in communication arising from tongue barriers. Digital communication can also increase efficiency and safety in traffic management, as passage details shared with a vessel traffic system and other ships would provide tools for improving the visibility of undertaken routes [42]. In areas not covered by the VTS, one of the vessels should take this role. The strategy related to this choice is one of the research topics of this paper.

Taking into consideration the above arguments and the results of our research on the BSA method for one-to-many encounter situations presented in [13], we did an extensive study to elaborate, develop, and implement the algorithm for solving multi-surface vehicle encounter situations. The main objective for collision avoidance algorithms was to keep a safe distance between the vessels. At the same time, we focused on optimizing the deviation distance covered by all surface vehicles involved in an encounter situation.

## 2. Generating of Anti-Collision Trajectories in a Multi-Surface Vehicle Encounter Situation (Many-to-Many Variants)

A many-to-many solution means that we consider the CPA value for each pair of surface vehicles and generate a set of safe trajectories for each surface vehicle in an encounter situation. The main idea of the proposed MBSA is to run a BSA [13] for each surface vehicle in the given sea area, in the order determined by the value of the risk function. The solution assumes we have full navigation data, such as the planned itineraries, courses, speeds, and minimal distance at the closest point of approach (DCPA), for each surface vehicle involved in a given many-to-many encounter situation. The MBSA result is a set of anti-collision trajectories given to ensure that at any point in the resulting trajectory the minimum DCPA for each pair of surface vehicles is not violated.

The first step in the MBSA is to order the surface vehicles by increasing the value of the collision risk. As stated in [43], the function for a collision risk assessment has only been defined for the one-to-one and one-to-many cases so far. This is natural, because there are not any systems that support many-to-many situations yet.

BSA spawns an anti-collision path, which comprises two or more segments of a polygonal line, warranting the safety condition (i.e., the value of the distance at the point of closest approach, DCPA). The algorithm is designed to choose the shortest path possible whilst maintaining a proper DCPA value. The resultant path is calculated based on the navigational information gathered from the perspectives of both “own” and “target” surface vehicles. One crucial feature of BSA is the fact that this method considers all types of collision scenarios, especially the most dangerous ones, such as the last moment maneuver [44].

The structure of the BSA permits the algorithm to solve the generic problem for safe and optimal sets of trajectories, not only for own surface vehicles but for all surface vehicles in the scenario. In the so-called many-to-many approach, navigational data are received from a group of surface vehicles in the given area (e.g., within a 30-nautical-mile radius), which are considered input data. The result is a set of anti-collision paths that do not violate the DCPA rule for any pair of surface vehicles and also have the lowest possible summed length. Such a generic version of the problem is key for the research and development of autonomous shipping technology. As far as global deployment is concerned, the software design for such remote decision-making units is very importance. Those solutions would generate a set of paths for the whole surface vehicle configuration and transmit those trajectories directly to the parties and surface vehicles involved in the scenario.

The authors proposed BSA, which is based on a beam search strategy [13], in order to determine a group of safe paths for an own surface vehicle. The beam search is a strategy used to search a tree of solutions by expanding only the most “promising” nodes (in terms of solution quality). It has proved successful in many optimization problems [45,46] and is a convenient heuristic to analyze a tree of trajectories (formed by maneuvers of surface vehicles). The BSA produces a set of CPA-safe trajectories (which are calculated by taking into account hard to port and hard to starboard maneuvers of own surface vehicles). At each step, a group of the most promising solutions is stored. They are then expanded via the addition of consequent actions. This algorithm has the following inputs:Designed itineraries N of target surface vehicles. Every itinerary is established using consecutive waypoints and can be handled as a polygonal chain (sections between consequent waypoints);Velocity and courses of all surface vehicles (it is presumed that surface vehicles change only their course, without altering their speed);The minimal value of DCPA applicable to all surface vehicles. The DCPA concept is described in the following text below;The maneuvering characteristics of the own surface vehicle, namely the maximal angle of turn b and minimal distance d between consequent actions. Both traits are dependent on the surface vehicle, and their values are set in such a way that the own surface vehicle should not significantly reduce its velocity in consecutive maneuvers (i.e., d is usually equal to 4–5 lengths of the surface vehicle).

In order to determine the risk of collision between the own surface vehicle and any enemy surface vehicle, the algorithm employs the DCPA and time to the closest point of approach (TCPA)measures. DCPA is the smallest distance achieved between two surface vehicles, if both of them maintain their current courses and speeds, whilst the time required to reach such DCPA is represented as the TCPA. The formulas are as follows:(1)DCPA=R·sinα
where *R* is the distance between two surface vehicles and α is the angle between the vector of relative speed $\left(V_{own}\to -V_{target}\to \right)$ and vector of relative position $\left(P_{target}\to -P_{own}\to \right)$. If α > 90 degrees, then DCPA = R.
(2)$$\mathrm{TCPA}=-\frac{{\mathrm{XV}}_{\mathrm{RX}}+{\mathrm{YV}}_{\mathrm{RY}}}{V_{R}^{2}}$$
where X and Y indicate the current relative position of the enemy surface vehicle, V_RX_ and V_RY_ describe the target relative velocity components, V_R_ is the relative velocity, and TCPA is the time to reach the closest point of approach [47].

In the MBSA, the following algorithm is proposed to determine surface vehicle ordering according to the decreasing value of collision risks in the many-to-many situations (Algorithm 1). It is simple but based on good sea practices.

The algorithm starts the trajectory planning from the surface vehicle with the greatest number of DCPA violations of other surface vehicles (IDCPA function) and continues with subsequent surface vehicles according to decreasing IDCPA values. If the IDCPA function is the same for two or more surface vehicles, the secondary criterion is used—total TCPA.
**Algorithm 1:** Sorting of Ships by Collision Risk1:  **for each** Ship(i) **do**
2:    IDCPA(i)=0; 3:              TCPA(i)=0;4:               **for each** Ship(j) **do**
5:        **if** (i≠j and Ship(i) violates DCPA(j) ) 6:       IDCPA(i)++; 7:               TCPA(i)=TCPA(i)+TCPA(i,j);8:  sort Ships(i) by IDCPA(i) according to the decreasing value first and, secondly

According to the decreasing value of TCPA(i), the MBSA runs the BSA for each ship(i) in order of surface vehicles, which is the result of Algorithm 1. The main steps are presented below as Algorithm 2.
**Algorithm 2:** Multi-ship Beam Search (MBSA) 1: Sort Ships by the collision risk (Algorithm 1)2. **for each** Ship(i)3:    CT: List current trajectories, where T is one trajectory in the list;4:    FT: Set of final trajectories (FT is initially empty)5:    OT: Ship(i) trajectory.6:    asT: Currently processed segment of T trajectory.7:    maxN: Maximal number of processed solutions.8:    Add OT to CT9:    **while** CT is not empty **do**10:      Remove T (the first trajectory of CT) from the start of list CT11:      **for each** Ship(j)’s trajectory **do**12:      **if** (i≠j) and the DCPA of the Ship(j) is violated in section as T by Ship(i)’s        trajectory **then**13:      Generate two anti-collision maneuvers (port and starboard) andsegments back to the waypoint14:      Based on the two maneuvers and T, create two new trajectories(T_1_ and T_2_)15:         **else**16:      Generate an artificial collision course T_A_ of the first ship withShips(j)’s trajectory.17:      For the artificial course T_A_, generate two anti-collision maneuvers(port and starboard) and segments back to the waypoint18:      Based on the two maneuvers and T_A_, create two new trajectories(T_1_ and T_2_)19:        **end if**20:      **for each** trajectory from {T_1_, T_2_} set **do**21:    **if** the first of two newly created segments of Ti trajectory is DCPA safe withrespect to all remaining ships **then**22:         **if** the segments are last segments in Ti and both are DCPAsafe **then**23:          Add Ti to FT24:         **else**25:          Add Ti to the end of list CT26:         **end if**27:       **end if**28:      **end for**29:    **end for**30:    **if** size(CT) > maxN **then**31:      Remove size(CT) from maxN worst trajectories(in terms of expected length) from CT32:    **end if**33:  **end while**34:  Choose the best trajectory (or trajectories) from FT.

The MBSA begins with the planned voyage of the first surface vehicle. Then, it checks whether the analyzed section of the path leads to a risk of collision (i.e., DCPA < minimal DCPA) with any vehicle in the target surface vehicles group, known as target(j).

Should such a situation arise, two novel paths are formed from the current path segment. To pass the target(i) at a safe distance at least meeting the minimal DCPA, one of those paths focuses on a starboard turn at a minimal angle, whilst the other includes a turn to the port side. Then, after target(i) is passed safely, yet another action is calculated, which involves turning towards the desired waypoint. The turning back point is computed using a binary search method for the closest coordinates, whereby the turning maneuver would not violate the DCPA rule for each target(i). Should the first anti-collision section be safe in regards to the interactions with every other target, the processed path is added to the group of solutions. The other segment responsible for turning back would be then analyzed in the next iteration. If the last waypoint can be safely reached, the whole path adjoins the outcome solution set.

When the first surface vehicle can safely pass target(i), the method generates an artificial collision course and two anti-collision paths, similar to those described in the previous paragraph. Such an approach helps to explore the solution space in a more effective way. There are some scenarios where only one action is required in order to pass a number of surface vehicles.

At every step, the algorithm accumulates no more than maxN best trajectories in terms of their expected length and then expands these by adding new maneuvers, enabling the first surface vehicle to pass any target at a correct distance. Consequently, maxN best trajectories are picked to be processed in the next iteration. The algorithm can, thus, be labeled as a beam search, as it explores a tree of correct and safe paths. Moreover, at every step it refills and substitutes the substandard trajectories with the best and most promising partial solutions.

The execution time of the MBSA is also of key importance, since it measures the execution time for one surface vehicle multiplied by maxN, such that the time does not exceed a few seconds, even with maxN = 30. The theoretical pessimistic time complexity is equal to O (maxN^3^). The average complexity is drastically lower because of the lower cases in which a full tree of possibilities of the actual part of trajectory is generated and the practical assumption that the area of 30 NM is considered as the area of many-to-many encounter situations.

## 3. The Results

We ran the MBSA in tests using the following assumptions concerning MBSA or BSA, with the main components of the former outlined below:The highest priority is given to safety measures, i.e., the DCPA chosen by the navigator should be kept at as high value as possible at all times;Only when the aforementioned conditions are met is the shortest route chosen;Multistage (numerous course alterations to reach the destination or the consequent waypoint) solutions are available;The route is updated on receipt of each novel AIS or ARPA data string;The next waypoint is 15 Nm from the current position;COLREGs are not implemented;The specific parameter of the BSA (described as maxN) is the maximum number of partial solutions that are considered within the tree structures computed by the algorithm. This variable is of utmost importance in regards to the quality of the result and the complexity of the algorithm. The higher the value of maxN, the closer the solution is to being optimal, albeit at the expense of longer computation time. During the tests, maxN was set as 1000;The BSA run time duration is shorter than 0.1 s on average. The algorithm is carried out every time the data is updated (with each AIS message, at least every 10 s). In order to mitigate frequent maneuvers, the algorithm attempts to maintain the current trajectory (or make only minor changes), unless the current path is not safe anymore or a much shorter and safer trajectory is created based on the updated locations.

The MBSA was tested in the situations presented in Figure 1, Figure 2, Figure 3 and Figure 4 (the unit is one meter and, in all situations, the minimal DCPA for all surface vehicles is equal to 1000 m). Numbered dots in trajectory lines indicate vehicle positions at subsequent time steps.

The main goal of the experiment was to check if the algorithm was correct, meaning that all anti-collision trajectories satisfy the condition of no DCPA violation at any point of each trajectory. Additionally, the length of each trajectory is checked in comparison to its straight length (assuming that there is no collision) between the planned waypoints.

In Figure 1, surface vehicle A is in a collision situation with B and C, while B and C are on parallel courses. The minimal DCPA is equal to 1000. The MBSA gives priority to surface vehicle A and calculates a trajectory that does not require surface vehicles B and C to alter their courses in this case.

In the situation presented in Figure 2, surface vehicle A is in a collision situation with three surface vehicles (i.e., B, C, and D). The maneuver suggested by the MBSA directs the surface vehicle astern of surface vehicles B and C, then ahead of surface vehicle D. In all cases, the DCPA value is above the limit. The results obtained in the situation presented in Figure 2 are similar to those in Figure 1 (only surface vehicle A is maneuvering).

The situation presented in Figure 3 is totally different from a navigational viewpoint. It is obvious that any action of any single surface vehicle will not solve all encounter situations. The collision avoidance results obtained by the MBSA were very promising. Although the navigational situation was complicated, not all surface vehicles had to execute a maneuver. For example, surface vehicle B managed to avoid collisions by maintaining its course and speed. In all cases, DCPA was at least equal at the assumed safe distance. Each course alteration was noticeable by other surface vehicles. What is worth underlining is that the execution of the maneuvers suggested by MBSA did not make three or more surface vehicles meet in one place.

The situation presented in Figure 4 is built on the simulation from Figure 3. An additional 14 vehicles are added to reach the limit of targets, which can be tracked by ARPA. It is obvious that in this case that many surface vehicles have to execute evasive actions. Finally, 11 of the vehicles could maintain their course and speed, while the remaining vehicles executed multiple waypoint routes to avoid collisions.

## 4. Discussion

We underline in the introduction that our research on the MBSA is dedicated to future software solutions for VTS and VTMS to support and manage semiautonomous and autonomous surface vehicle traffic. Therefore, we assume that such external IT systems (with MBSA as the main function) have full control over surface vehicles located in an open sea area and are used in the first phase (as early as possible) of many-to-many encounter situations. In other words, each surface vehicle in a many-to-many encounter situation is a slave surface vehicle and MBSA is a master-type remote center (the artificial brain for surface vehicles). Only when an external risk occurs as defined in the control system does MBSA stop.

The results of the experiment are very promising, because none of the resulting anti-collision trajectories violate minimal DCPA along the entire length of the trajectory. This is very important, as it confirms the hypothesis that the central management of surface vehicles by remote centers (future VTS and VTMS) assures the security of the whole system and the effectiveness of the anti-collision routes.

To verify our assumptions and the algorithm, we used examples from [13], in addition to the extremely difficult collision situations presented in Figure 3 and Figure 4. These cases were designed to meet a few assumptions. First of all, the encounter situation cannot be solved by maneuvers by a single surface vehicle. The different situations include head-on, crossing, and overtaking scenarios. For example, in Figure 3 surface vehicle A has to deal with an opposite surface vehicle, overtaking surface vehicle, and three crossing surface vehicles. Finally, most of the surface vehicles taking part have one meeting point, with DCPA being equal to zero. The abovementioned factors make this encounter situation critical. The results obtained using MBSA are very promising, and importantly are considered acceptable by experienced navigators—namely surface vehicle captains.

Surface vehicle A altered its course to the starboard to pass astern of surface vehicles E and F. Using this maneuver, it also managed to avoid a collision with surface vehicle B. In this way, surface vehicle B could maintain its course and speed. Surface vehicle C altered course to port in order to pass astern of surface vehicles A and D. This maneuver also allowed surface vehicle C to avoid a close-quarter situation with surface vehicle F. Surface vehicle D, overtaking surface vehicle A, altered its course to port, and using only one maneuver was able to pass ahead of surface vehicles C, E, and F. At the same time, it overtook A and avoided a collision with surface vehicle B going south. Surface vehicles E and F altered course to starboard, and with only one maneuver were able to avoid all close-quarter situations. If we imagine that all surface vehicles have altered their courses to starboard, then surface vehicle A would need to alter its course by at least 45 degrees. This is why surface vehicle D following A would need to turn even more to starboard. If surface vehicle C changed to starboard, it would be in close quarters with surface vehicles F and B and on a parallel course with surface vehicle E, meaning the situation is not solved. To summarize, the presented solution is safe and effective.

In the situation presented in Figure 4, there are 20 vessels involved. In an area measuring 15 by 20 km, a complicated collision situation was simulated. Surface vehicles were involved in all types of encounter situations, i.e., overtaking, crossing, and head on scenarios, with particular emphasis on overtaking, which usually generates more challenges than other situations as it takes a longer time. The MBSA calculated trajectories for all vehicles, enabling them to avoid all collisions and additionally enabling them to pass all other targets at the safe presumed distance of 1000 m. The simulation results can be viewed in [48].

The same situation as for BSA occurs for MBSA, whereby the COLREGs are not implemented in MBSA [13]. However, it is possible to change the MBSA to execute the first course alteration to starboard. In this situation, the probability of generating length-optimal trajectories will decrease. It is worth emphasizing that the safe DCPA value along the entire length of the anti-collision trajectory is the most important and inviolable condition of the MBSA method. Additionally, the condition of the shortest possible length of the trajectory is considered [13].

To the best of our knowledge, the MBSA is the first anti-collision algorithm that solves the collision problem for many-to-many encounter situations; we assume that the algorithm will be run in the future in remote navigation integrated within systems designed for autonomous surface vehicles. Therefore, we prepared benchmark data (Table 1) for the situation presented in Figure 3 for comparison between the MBSA and other solutions.

## 5. Conclusions

We propose the novel and original MBSA, which solves the collision problems in many-to many encounter situations in the open sea. We conducted a series of simulations and experiments, which confirmed that each of the resulting trajectories in each examined situation is fully safe—the value of the minimal DCPA is never exceeded for any pair of surface vehicles. Moreover, the lengths of the resultant trajectories are shorter than those generated by commercial systems such as NAVDEC, which are executed for each ship in a one-to-many encounter situation [13]. These conclusions let us assume that the MBSA can be implemented in future remote-control systems used in VTS or VTMS centers.

The goal was to find at least one trajectory for each vehicle that enabled them to pass all other vehicles at a safe distance, taking into account all vehicles when planning the evasive action and avoiding the situation when the action planned in relation to one target is dangerous to another target.

The situations presented in Figure 3 and Figure 4 are rather unusual, very complicated, and not easy to solve. The solutions calculated by the MBSA do not follow COLREGs, however they fulfil the assumed safety parameters, i.e., safe distances between all targets. As a result, each vehicle avoided collision.

## 6. Future Work

Our future work will concentrate on the following directions and goals:

Goal No. 1:

As MBSA has a modular structure, we can replace the risk assessment function with another one that is more specific in the future, considering additional parameters such as the dynamic characteristic of the surface vehicle. We plan to define, based on a literature review [43], different versions of collision risk functions and examine how each of those functions influences the quality of MBSA results in terms of the distant efficient anti-collision trajectories (execution time and number of course alterations).

Goal No. 2:

One issue to be considered is ship dynamics, which reflect real conditions. An interesting approach regarding ship dynamics is presented in [49], where the trajectory generation problem is divided into separate stages and ship dynamics are solved in the last stage. Our future research will concentrate on a method for incorporating ship dynamics in MBSA.

Goal No. 3:

We assume that only one area of many-to-many encounters is covered by the MBSA operation. The size of this area is 30 NM for the purpose of our experiments. We are aware that we must design the next version of the algorithm to be capable of covering the whole open sea area monitored and managed by future VTS or VTMS centers.

Goal No. 4

We conducted our experiments using only simulations. We want to test the MBSA on real-life data, which we hope to possess thanks to close cooperation with Polsteam [50]. We cannot plan any real-life experiments, but we are able to prepare and perform our experiments using the MBSA in quasi-real conditions, thanks to the Foundation for Safety of Navigation and Environment Protection in Iława (in Poland) [51], which has prepared semiautonomous and autonomous models of ships through the implementation of the AVAL R&D project [10]. Two of the authors are working as scientists on this project [52].

## Figures and Tables

**Figure 1 sensors-20-04115-f001:**
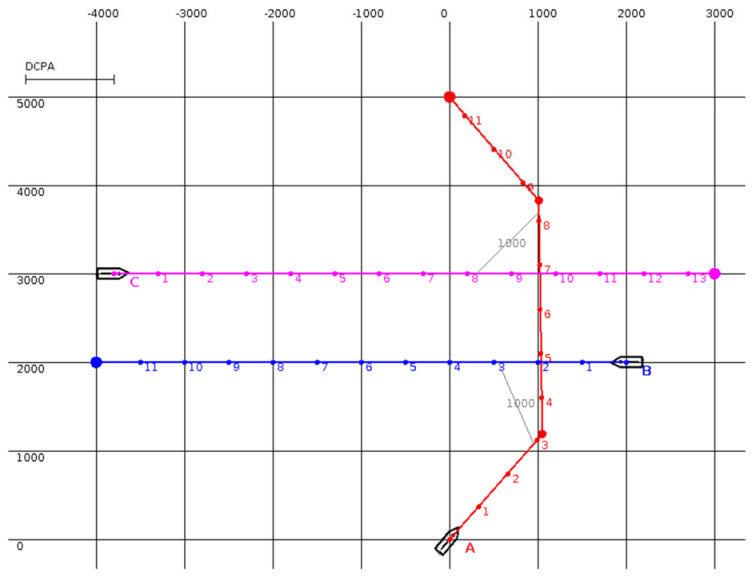
An encounter situation involving three surface vehicles (A, B, and C). The trajectories of surface vehicles B and C are marked in blue and pink, respectively. The anti-collision trajectory for surface vehicle A, generated by MBSA, is marked in red.

**Figure 2 sensors-20-04115-f002:**
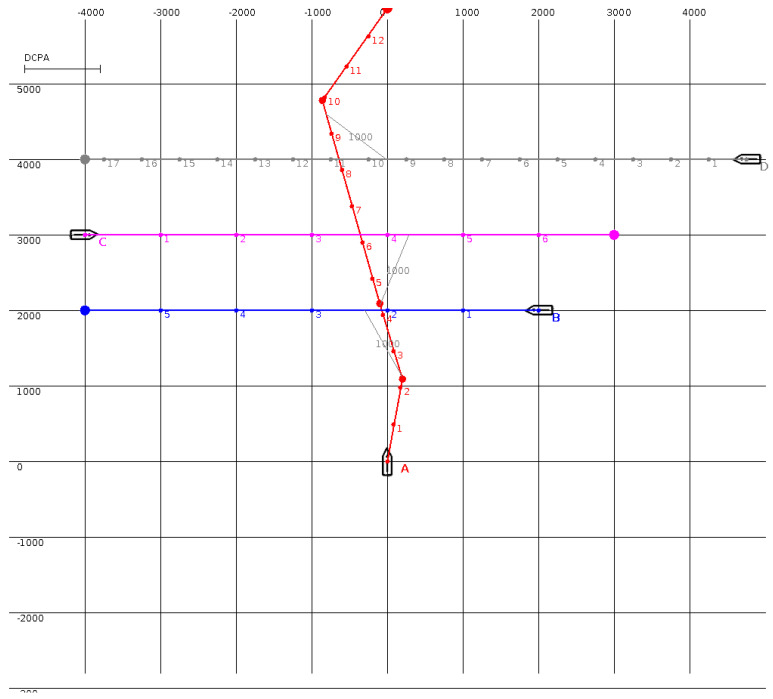
An encounter situation involving four (A, B, C, and D) surface vehicles. The trajectories of target surface vehicles B, C, and D are marked accordingly in blue, pink, and grey. The anti-collision trajectory for A surface vehicle proposed by the MBSA is marked in red.

**Figure 3 sensors-20-04115-f003:**
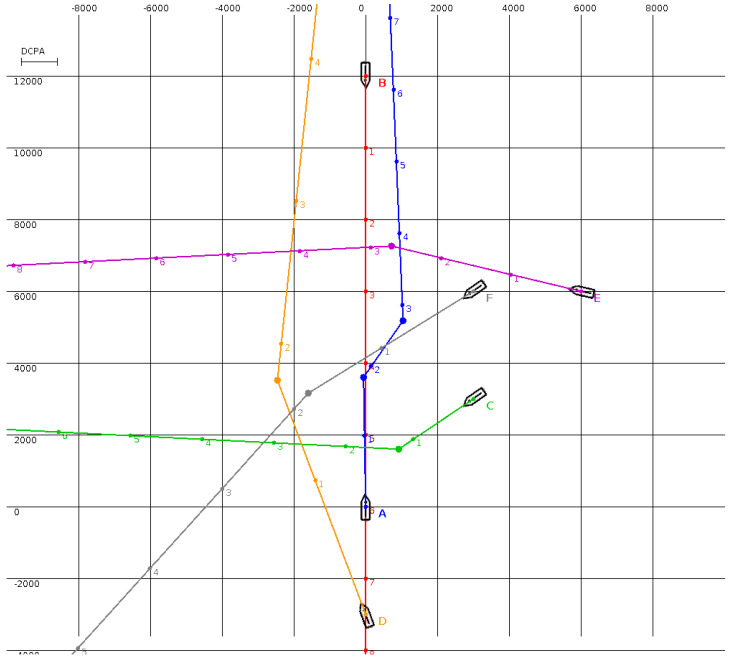
A multi-surface vehicle encounter involving 6 vehicles. The initial positions are marked by surface vehicle contours. The initial course for vehicles A and D is 000, for vehicle B is 180, for vehicles C and E is 270, and for vehicle F is 225. The anti-collision trajectories for each surface vehicle generated by the MBSA are presented in different colors.

**Figure 4 sensors-20-04115-f004:**
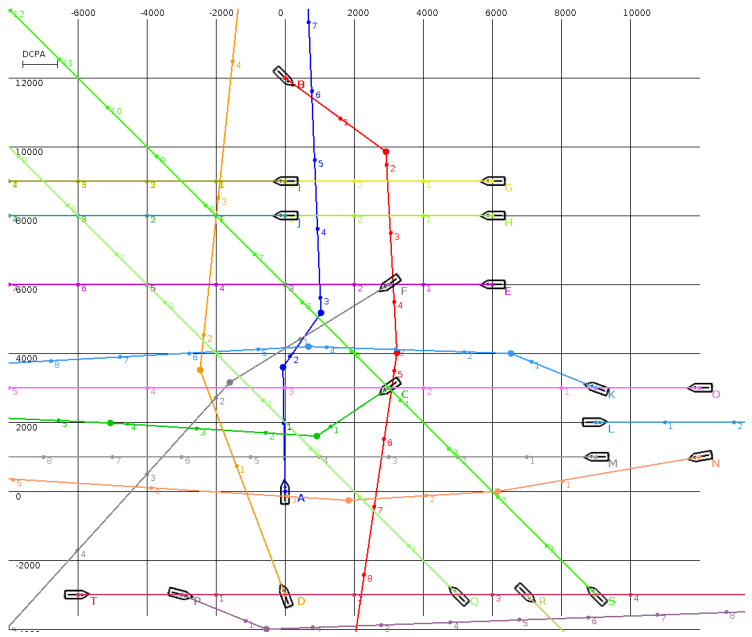
A multi-surface vehicle encounter involving 20 vehicles.

**Table 1 sensors-20-04115-t001:** Benchmark data for presented simulations

Surface Vehicle	Start Point	End Point	The Length of Planned Trajectory	The Length of the Anti-Collision Trajectory	The Increase in Percentage of the Length of the Trajectory
A	(0, 0)	(0, 30,000)	30,000	30,385.7	1.286%
B	(0, 12,000)	(0, −18,000)	30,000	30,000	0
C	(3000, 3000)	(−27,000, 3000)	30,000	30,458.4	1.528%
D	(0, −3000)	(0, 27,000)	30,000	30,582.3	1.941%
E	(6000, 6000)	(−24,000, 6000)	30,000	30,182.7	0.609%
F	(3000, 6000)	(−18,213, −15,213)	30,000	30,179.1	0.597%

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
