# Peer review of "Beam Search Algorithm for Anti-Collision Trajectory Planning for Many-to-Many Encounter Situations with Autonomous Surface Vehicles"

_sensors, 2020, doi:10.3390/s20154115_

Round 1

Reviewer 1 Report

This paper presents the idea of dealing with multiple encounters using the Beam heuristic search algorithm that explores a graph by expanding the most promising node in a limited set. But in beam search, only a predetermined number of best partial solutions are kept as candidates.

Major comments:

  • COLREGs suggest that the ship should respond to a collision scenario by evasive actions corresponding to one-by-one scenarios
  • The method presented violates COLREGs
  • The “safe” related parameters are not reliable and sustainable for collision avoidance.

Minor comments:

  • The use of TCPA and TCPA separately make no sense in terms of defining collision risk. In real conditions, TCPA and TCPA can reflect the encounter situation, but the variation of a parameter (TCPA) can not be used to define the collision risk level. 
  • Relative distance and DCPA are different concepts. The safe distance can not be determined using minimal DCPA, which is unreliable.
  • The research ignores maneuverability.
  • The authors presume that the ship's speed is stable, which is not realistic as it may lead to meaningless  DCPA and TCPA estimates 
  • The simulation settings are too simple to simulate collision avoidance. They cannot be applied in port areas, waterways, or high ship density areas.
  • It is evident that the method will fail in overtaking scenarios because the DCPA is stable.
  • There is a lack of explanation of the collision risk assessment analysis used in the study.
  • An explanation of why the authors did these various experiments using real scenarios should be provided.

Data-driven collision avoidance method should be the preffered research direction under commonly agreed navigation standards, rather than simulation.

Author Response

Major comments:

  • COLREGs suggest that the ship should respond to a collision scenario by evasive actions corresponding to one-by-one scenarios
  • The method presented violates COLREGs
  • The “safe” related parameters are not reliable and sustainable for collision avoidance.

Authors use DCPA and TCPA as parameters to assess the risk of collision. From the author's personal experience, those parameters are widely used on ships to assess risk of collision. Also ARPA, which is a very popular tool supporting navigator decision in collision situations, uses those parameters to assess the risk and to suggest evasive actions (Trial manoeuvre). Executing evasive action based on DCPA 1 Nm and TCPA 15 minutes, enables safe passage in one-to-one situations.

Minor comments:

  • The use of TCPA and TCPA separately make no sense in terms of defining collision risk. In real conditions, TCPA and TCPA can reflect the encounter situation, but the variation of a parameter (TCPA) can not be used to define the collision risk level. 

Authors use DCPA and TCPA as parameters to assess the risk of collision. Evasive action is taken, when at least on target does not fulfil safety criteria i.e. DCPA and TCPA simultaneously. In the crowded situation e.g. fig. 3, usually evasive action in relation to one dangerous target, creates a dangerous situation with others. The Author's goal was to find at least one trajectory for each vehicle, that enables to pass all other vehicles at the safe distance. Taking into account all vehicles when planning evasive action, avoid the situation, when action planned in relation to one target is dangerous to another target. The challenge was: which vehicle should be first. Only for this purpose, we have generated parameters, which is the sum of DCPA’s. The vehicle with the smallest sum of DCPA’s is considered as the first to calculate. However this calculation starts, when both parameters i.e. DCPA and TCPA are not met.

  • Relative distance and DCPA are different concepts. The safe distance can not be determined using minimal DCPA, which is unreliable.

There are numerous approaches to collision assessment. The most common and regarded as inseparably linked to good seamanship is filtering by DCPA and TCPA. Depending on a navigator, a ship’s domain can also be introduced, describing the minimal passing distance taking into account aspects of the approach. In such cases there is a general tendency to keep CPA as high as possible while passing in front of enemy’s ship transverses, and decrease applicable DPCA when passing behind the vessel’s midships. Authors agree that relative distance and DCPA are completely different concepts. The study is based on the premise that as long as own ship keeps requested safe DCPA, there should be no dangerous situations resulting in abnormally close encounters. How the safe DCPA is determined is a completely different, not to mention broad, topic. In the presented article DCPA was a fixed value that could be altered for any simulation cycle. There is a discussed topic among researchers on introducing a ship domain rather than circular-like domain in the form of fixed DCPA, but then again it would be presented in studies yet to be conducted.

  • The research ignores maneuverability.

As stated within the second section of the paper, the algorithm examines the maneuvering characteristics of the own surface vehicle, namely, the maximal angle of turn b and minimal distance d between consequent actions. Those both traits are dependent on the surface vehicleship, and their values are set in such a way that the own surface vehicle should not reduce significantly velocity in consecutive maneuvers (i.e., d is usually equal to 4–5 lengths of the surface vehicle).

  • The authors presume that the ship's speed is stable, which is not realistic as it may lead to meaningless  DCPA and TCPA estimates 

The course alterations were restricted by the maximum value of 60 degrees, whilst the distance between consequent maneuvers is designed to be about four vehicle lengths. The algorithm always chooses the shortest path possible that meets the criteria. Said path is to suggest as little course deviation as possible. Based on numerous marine vehicle maneuvering data it can be assumed that course changes up to 30 degrees does not impact the travel time in any significant way.

  • The simulation settings are too simple to simulate collision avoidance. They cannot be applied in port areas, waterways, or high ship density areas.

The Authors fully agree that the presented algorithm is not dedicated to port areas. As stated in conclusions and mentioned throughout the entirety of the paper the algorithm is to be used on the open seas.

  • It is evident that the method will fail in overtaking scenarios because the DCPA is stable.

The purpose of this article is to show the effectiveness of the algorithm which bases on the circular vessel’s domain. Should the experiment implement different domain shapes, such as elliptical or polygonal, the applicable passing distance, particularly during overtaking, could be lower.

  • There is a lack of explanation of the collision risk assessment analysis used in the study.

The risk of collision stems from good seamanship and is determined as a simultaneous value of DCPA less than 1.0 Nm and TCPA less than 15 minutes.

  • An explanation of why the authors did these various experiments using real scenarios should be provided.       

Due to high costs of real testing scenarios rendering such instances scarce, in this study virtual simulation was picked as a validation method. The high complexity of the situation has better outlined the algorithm in terms of the quality of the results obtained. Thanks to that its application in real conditions should be in general correct, because, as a rule, factual navigational circumstances tend to be simpler situations.

Data-driven collision avoidance method should be the preferred research direction under commonly agreed navigation standards, rather than simulation.

Authors of the paper agree that there should be commonly agreed navigational standards regarding the research direction. Yet, due to the high costs of ship rental, the real testing scenario is not always available. In a previous study and associated article (solution one-to-many), it was possible to test the algorithm on board ships of Polsteam, only due to the shipowner's benevolence. This time however it was not a case, as it was not possible to engage so many vessels in the testing procedure..

Reviewer 2 Report

The title of the article is very general and does not correspond to its content, which is confirmed by the authors in the conclusion of 4 goals to be developed in the future, so that all this has any practical sense.
1. I propose to complete the title of the article to the following form: "Beam Search Algorithm for Anti-Collision Trajectories Planning for Many-to-Many Encounter Situations of Autonomous Surface Vehicles".
2. Complete the Abstract with the sentence: "The article specifies the problem of anti-collision trajectories planning in many-to-many encounter situations of moving autonomous surface vehicles excluding COLREGs Rules and vehicle dynamics". Also state what devices are the source of information for the algorithm, or just VTS, as the authors state in the end, or also the ARPA radar system.
3. In Keywords instead of "autonomous ship" use "autonomous surface vehicle".
4. Use "surface vehicle" instead of "ship" or "vessel" throughout the article because of the omission of COLREGs and its dynamics.
5. In section 2, on page 5 and later, the concept of collision risk is used as a function of DCPA and TCPA. Please, present this function as an appropriate mathematical formula.
6. It should be noted that the risk of collision also depends on the distance between ships, please take this into account.
7. Providing formulas (1) and (2) does not make sense if you use ready information from VTS and ARPA.
8. At the bottom of page 8 and further in the drawings no units are given, for example "in all situations minimal Dcpa for all ships was equal 100".
9. Complete the article with the results of studies of the passing situations of more vehicles, for example 20, which can track the ARPA system.
10. Conclusions require completely different content, corresponding to the content of the article, and not the futuristic wishes of the authors, not supported by any research results.

Author Response

     The authors would like to thank all the reviewers for their comments and suggestions. We do appreciate your inputs, which help us to improve the content.

The revised manuscript was prepared in such a way, that all new parts were underlined to make the revision process as easy as possible.

The proofreading of the article was done.

  1. I propose to complete the title of the article to the following form: "Beam Search Algorithm for Anti-Collision Trajectories Planning for Many-to-Many Encounter Situations ".

Thank you for the suggestion. The proposal was introduced in the article.

Thank you for the suggestion. The proposal was introduced in the article. Information about the sources of data (ARPA and simulation) was added in the Abstract.

Thank you for the suggestion. The proposal was introduced in the article. 

Thank you for the suggestion. The proposal was introduced in the article.

  1. In section 2, on page 5 and later, the concept of collision risk is used as a function of DCPA and TCPA. Please, present this function as an appropriate mathematical formula.

Collision risk function has been presented as an mathematical formula.

  1. It should be noted that the risk of collision also depends on the distance between ships, please take this into account.

The distance between ships is relative in the parameter TCPA.

Authors use DCPA and TCPA as parameters to assess the risk of collision. Evasive action is taken, when at least on target does not fulfil safety criteria i.e. DCPA and TCPA simultaneously. In the crowded situation e.g. fig. 3, usually evasive action in relation to one dangerous target, creates a dangerous situation with others. The Author's goal was to find at least one trajectory for each vehicle, that enables to pass all other vehicles at the safe distance. Taking into account all vehicles when planning evasive action, avoid the situation, when action planned in relation to one target is dangerous to another target. The challenge was: which vehicle should be first. Only for this purpose, we have generated parameters, which is the sum of DCPA’s. The vehicle with the smallest sum of DCPA’s is considered as the first to calculate. However this calculation starts, when both parameters i.e. DCPA and TCPA are not met.

  1. Providing formulas (1) and (2) does not make sense if you use ready information from VTS and ARPA.

Mentioned formulas are well known. Authors use them to assess risk of collision. This risk can be assessed based on VTS and ARPA information. We fully agree. Mentioned formulas are used also to generate safe trajectories. Such functionality is usually not available in VTS or ARPA. This is why we have added those formulas. We are open to remove them in the second round of evaluation.

  1. At the bottom of page 8 and further in the drawings no units are given, for example "in all situations minimal Dcpa for all ships was equal 100".

Unit is one meter. In all drawings minimal DCPA has been unified to 1000 units (one kilometer).

  1. Complete the article with the results of studies of the passing situations of more vehicles, for example 20, which can track the ARPA system.

The collision situation presented in figure 3 is already unusual and not easy to solve. Authors, in the original paper, did not present situations with more targets to keep clearness of the figure. The situation with 20 autonomous surface vehicles is presented in figure 4. The movie presenting simulation can be viewed here: https://chilan.com/mbsaships20.mp4

  1. Conclusions require completely different content, corresponding to the content of the article, and not the futuristic wishes of the authors, not supported by any research results.

The conclusion part was complemented. The future work part was separated from the conclusion.

Round 2

Reviewer 1 Report

In my opinion the paper quality is improved. I understand the motivation and the responses by the reviewers show good effort. However, the in depth updates to the paper are limited, the quality of the scientific background deems improvement and references / literature survey should be explained.

On this basis I have to suggest major corrections. 

Author Response

Comments and Suggestions for Authors

In my opinion the paper quality is improved. I understand the motivation and the responses by the reviewers show good effort. However, the in depth updates to the paper are limited, the quality of the scientific background deems improvement and references / literature survey should be explained.

The authors appreciate the confirmation, that the article is improved. We would like to thank you for your feedback, which helps us to introduce improvements. We did a deeper analysis in the literature. Six new articles were added. They present our scientific background (Beam Search Algorithm), other methods of solving collision situations (PGHAPF method) as well as a different approach to the safe area around own ship (non-circular domain approach).

We are open on the other research methods and definitely will introduce them in our ongoing articles. It is difficult to change design deeper in this article as it is focused on a specific method. However, we are ready to redesign following your detailed suggestions.

On this basis I have to suggest major corrections. 

Reviewer 2 Report

Since the authors have taken into account all my comments and corrected the text of manuscript, I suggest accepting the article for publication.

Author Response

Since the authors have taken into account all my comments and corrected the text of manuscript, I suggest accepting the article for publication.

Thank you once again for your valuable comments and suggestions, which help us to improve the article.